# Estimated standard values of aerobic capacity according to sex and age in a Japanese population: A scoping review

**Hiroshi Akiyama**[1], **Daiki Watanabe**[2], **Motohiko Miyachi**[2]*

1 Graduate School of Sport Sciences, Waseda University, Tokorozawa, Saitama, Japan, 2 Faculty of Sport Sciences, Waseda University, Tokorozawa, Saitama, Japan

* miyachim@waseda.jps

**Data Availability Statement:** All relevant data are within the paper and its Supporting Information files.

## Abstract

Aerobic capacity is a fitness measure reflecting the ability to sustain whole-body physical activity as fast and long as possible. Identifying the distribution of aerobic capacity in a population may help estimate their health status. This study aimed to estimate standard values of aerobic capacity (peak oxygen uptake $[\dot{V}O_2\text{peak}]/\text{kg}$ and anaerobic threshold [AT]/kg) for the Japanese population stratified by sex and age using a meta-analysis. Moreover, the comparison of the estimated standard values of the Japanese with those of other populations was performed as a supplementary analysis. We systematically searched original articles on aerobic capacity in the Japanese population using PubMed, Ichushi-Web, and Google Scholar. We meta-analysed $\dot{V}O_2\text{peak}/\text{kg}$ (total: 78,714, men: 54,614, women: 24,100) and AT (total: 4,042, men: 1,961, women: 2,081) data of healthy Japanese from 21 articles by sex and age. We also searched, collected and meta-analysed data from other populations. Means and 95% confidence intervals were calculated. The estimated standard values of $\dot{V}O_2\text{peak}/\text{kg}$ (mL/kg/min) for Japanese men and women aged 4–9, 10–19, 20–29, 30–39, 40–49, 50–59, 60–69, and 70–79 years were 47.6, 51.2, 43.2, 37.2, 34.5, 31.7, 28.6, and 26.3, and 42.0, 43.2, 33.6, 30.6, 27.4, 25.6, 23.4, and 23.1, respectively. The AT/kg (mL/kg/min) for Japanese men and women aged 20–29, 30–39, 40–49, 50–59, 60–69, and 70–79 years were 21.1, 18.3, 16.8, 15.9, 15.8, and 15.2, and 17.4, 17.0, 15.7, 15.0, 14.5, and 14.2, respectively. Herein, we presented the estimated standard values of aerobic capacity according to sex and age in a Japanese population. In conclusion, aerobic capacity declines with ageing after 20–29 years of age. Additionally, aerobic capacity is lower in the Japanese population than in other populations across a wide range of age groups. Standard value estimation by meta-analysis can be conducted in any country or region and for public health purposes.

## Introduction

Aerobic capacity is a fitness measure that reflects the ability to sustain whole-body physical activity as fast and long as possible. Aerobic capacity is assessed by physiological indicators:

**Funding:** This work was supported by the Japan Science and Technology Agency, SPRING (Grant Number JPMJSP2128 to Hiroshi Akiyama) and Practical Research Project for Lifestyle-related Diseases Including Cardiovascular Diseases and Diabetes Mellitus from the Ministry of Health, Labour and Welfare (Grant Number 21FA1004 and 22FA1004 to Motohiko Miyachi). The funders had no role in study design, data collection and analysis, decision to publish, or preparation of the manuscript.

**Competing interests:** The authors declare that they have no competing interests.

**Abbreviations:** VO2max, maximal oxygen uptake; VO2peak, peak oxygen uptake; AT, anaerobic threshold; 95% CI, 95% confidence interval.

maximal oxygen uptake ($\dot{V}O_2max$), peak oxygen uptake ($\dot{V}O_2peak$), and anaerobic threshold (AT) [1–4]. $\dot{V}O_2max$ is defined as the maximum observable oxygen uptake at which no further increase in oxygen uptake occurs despite increased exercise intensity [5]. $\dot{V}O_2peak$, the highest value of oxygen uptake achieved during an incremental exercise test, is widely used as a proxy indicator [6,7]. Wasserman et al. defined AT as the exercise intensity or oxygen uptake just before the initiation of anaerobic metabolism, causing an increase in blood lactate and hydrogen ion concentrations and/or breath gas parameters [4,8–10]. In contrast to $\dot{V}O_2max$ and $\dot{V}O_2peak$, AT assessment does not require exercise up to exhaustion; thus, it has the advantage of minimising the burden on the participant and being unaffected by the participant's motivation [4,8].

The American Heart Association has stated the need to establish a global standard of aerobic capacity ('cardiorespiratory fitness' in the original) and emphasised the importance of regular assessment of aerobic capacity as clinical vital signs [11–13]. Several studies have reported that aerobic capacity is a strong predictor of all-cause and disease-specific mortality [12,14–19] and have recommended maintaining or improving aerobic capacity [20]. To establish more realistic and practical reference values, there is no controversy about the need to understand the current status of the population, that is, to obtain standard values of aerobic capacity.

The Japanese Ministry of Health, Labour and Welfare (MHLW) established a reference value for aerobic capacity for health promotion based on substantial epidemiological evidence regarding the association between aerobic capacity and risk of disease or mortality (1st ed. published 2006, 2nd ed. published 2013) [21]. However, establishing reference values based on standard values (mean and standard deviation [SD]) for the healthy population is warranted [22] because the reference values should consider both the epidemiological evidence and the physiological background in the target population and feasibility. Although difficult in practice, the standard values should be identified by directly measuring aerobic capacity by sex and age from a large number (thousands or more) of randomly selected healthy individuals from the target population and assessing their mean and distribution.

The MHLW reference values for aerobic capacity were set based on a systematic review of epidemiological evidence, including much data from other countries, without comparing the levels of aerobic capacity in the Japanese and other populations. The classic review by Shvartz and Reibold (1990) implied that $\dot{V}O_2peak/kg$ is lower in the Japanese than in other populations [23]. This observation underscores the necessity to compare the aerobic capacity between the Japanese and other populations. Suzuki et al. previously reported the reference interval of $\dot{V}O_2peak/kg$ for the Japanese population [24,25]; nevertheless, they could not compare the aerobic capacity between the Japanese and other populations due to the absence of data from other populations.

Therefore, the primary aim was to integrate data from previous studies to estimate the standard values of aerobic capacity by sex and age in the Japanese population and to map them systematically. The secondary aim was to compare the estimated standard values of the Japanese with those of other populations.

## Materials and methods

We conducted a scoping review, a methodical approach to literature analysis that effectively enables enhanced understanding of a specific focused topic through identifying and addressing knowledge gaps. The present scoping review was based on the PRISMA extension for scoping reviews (PRISMA-ScR) checklist, consisting of a five-step process. (1) identify research questions, (2) identify relevant articles, (3) select articles, (4) map the data, and (5) collate,

summarize, and report results. The PRISMA Scoping Reviews (PRISMA-ScR) checklist is shown in S1 File (S1 Checklist) [26,27]. The present study was exempted from review by the Waseda University Ethics Review Board because it is a literature study.

## Search strategy

We systematically searched and meta-analysed articles for the Japanese population from 1950 to 2023 using PubMed and Ichushi-Web on January 1, 2023. We aimed to estimate standard values of aerobic capacity in the Japanese population; thus, we attempted to obtain data from articles considering the populations as far as possible (i.e. descriptive study). The search terms consisted of relevant terms, such as 'Japanese', 'aerobic capacity', 'exercise test', and 'normal values' (S1 Table in S1 File). Medical Subject Headings were also employed to ensure that potential articles were not missing from the systematic search. The grey literature search was conducted using the Google Scholar database to identify other articles published in English or Japanese that could not be located by the systematic search. A grey literature search is the process of identifying informal and hard-to-obtain publications such as conference papers and reports that are not widely available through traditional channels for research purposes. The grey literature was searched throughout the writing of the manuscript and was retrieved until January 1, 2023. These methods helped to collect a broad range of valuable Japanese articles that could not be adequately covered by systematic search alone. Moreover, we also conducted an umbrella review for the secondary aim, that is, to compare the estimated standard values of aerobic capacity in the Japanese and other populations. An umbrella review is an analysis that integrates multiple reviews and meta-analyses on a specific topic and presents a comprehensive conclusion of the existing evidence.

The search strategy for the other populations is shown in the Supplemental data (S2 Table in S1 File).

## Eligibility criteria

The eligibility criteria included studies that reported aerobic capacity according to sex and age. The aerobic capacity indicators were $\dot{V}O_2$max, $\dot{V}O_2$peak, and AT. Generally, $\dot{V}O_2$max and $\dot{V}O_2$peak are different concepts that should not be confused [6,7]. However, in this study, we used studies that described both $\dot{V}O_2$max and $\dot{V}O_2$peak. The reasons for this are as follows. There were usually no differences between $\dot{V}O_2$max and $\dot{V}O_2$peak values, and if there were differences, they were small. A standard value estimated from $\dot{V}O_2$max alone data has a limited range of applications. The data, including $\dot{V}O_2$peak, can be generalised to individuals who symptomatically stopped the exercise test, that is, individuals who cannot be pushed to their physiological limits. It was difficult to decide between $\dot{V}O_2$max and $\dot{V}O_2$peak because much of the article does not adequately explain how to distinguish between the concepts of $\dot{V}O_2$max and $\dot{V}O_2$peak. $\dot{V}O_2$max requires several criteria to be met, including levelling off, and is part of the observed value of $\dot{V}O_2$peak. Therefore, in the present study, the standard values were expressed as the standard values of $\dot{V}O_2$peak. In addition, the present meta-analysis included data from both direct methods (measured using breath gas analysis during exercise test) and indirect methods (estimated using heart rate at several specific exercise intensities during submaximal exercise).

The inclusion criteria were as follows: (1) human studies; (2) studies measuring $\dot{V}O_2$max, $\dot{V}O_2$peak, or AT by exercise tests; (3) studies with a study period between 1 January 1950 and 1 January 2023; (4) studies in which data were reported by sex and age; and (5) studies in which the exercise mode was cycling or running, because large muscle groups are used and the

amount of external load is easy to control. The exclusion criteria were as follows: (1) studies in which data were not reported by sex and age; (2) studies not in Japanese or English; (3) studies of individuals with serious diseases, such as angina pectoris, acute myocardial infarction, and chronic obstructive pulmonary disease; (4) studies for which the full text was not available despite requests for articles from the first author and library services; and (5) studies in which data were duplicated (in which case, the study with the more appropriate sample size/reported information was selected).

The eligibility criteria for the other populations are shown in the Supplemental data.

## Study selection and data extraction

In primary screening, papers potentially containing information on the aerobic capacity of Japanese by sex and age were selected from titles and abstracts by two independent researchers (H.A. and M.M.). In the secondary screening, the same independent researchers perused the full text of the papers selected in the primary screening and selected those that precisely met the eligibility criteria. The papers handled during selection were managed using Mendeley Desktop (version 1.19.8) by the two researchers. From the papers selected in the secondary screening, data on (1) the first author's name, (2) year of publication, (3) sex, (4) age, (5) exercise mode, and (6) means and distributions (SD, standard error, or confidence interval [CI]) of $\dot{V}O_2peak/kg$ and the value of AT/kg were extracted. The '/kg' here means weight correction, not lean mass. If numerical data were not reported in the included study, we enquired about the numerical data from the corresponding author of the study. If the corresponding author did not respond, numerical data were extracted using WebPlotDigitizer version 4.5 (Ankit Rohatgi, Pacifica, CA, USA) [28] when figures were included in the study. In case of disagreements among the investigators regarding the extracted data, a final decision was made through discussion until a consensus was reached.

The study selection for the other populations is shown in the Supplemental data.

## Data analysis

Statistical analyses were performed using Microsoft Excel for Windows (version 2206). We calculated $\dot{V}O_2peak/kg$ (partly $\dot{V}O_2max/kg$) and AT/kg according to sex and age from the included studies by the simple mean and SD to avoid the impact of studies with a large sample size. In particular, the sample size of a study by Kono et al. (1997) was very large, accounting for 71% (55,521/78,714) of the total sample size in the present meta-analysis, which may affect the mean and SD [29]. In addition, Kono et al. estimated $\dot{V}O_2peak/kg$ by an indirect method with a submaximal exercise test. Considering these factors, we used simple mean and SD instead of weighted values in the present meta-analysis.

All statistical analyses were stratified by sex and age group to estimate the standard values of aerobic capacity for the Japanese and other populations. Age group stratification was calculated using simple mean and 95% CI for each 10-year age group. Scatter plots were created using data on each aerobic capacity and age, and their correlations were examined. Their relationship is influenced by growth [30] and ageing [31]. Then, the slopes and intercepts of the association of age with $\dot{V}O_2peak/kg$ and AT/kg for each age category ($\leq$19 years, $\geq$20 years) were determined using a linear approximation model with the least squares method to consider these influences. The two age categories in both the Japanese and other populations were classified according to the Japanese adult criteria (legal age). Consequently, linear functional equations were developed to predict age-related changes in each aerobic capacity by sex in the Japanese and other populations. The coefficient of determination ($R^2$) was used to assess how

well the regression model explains the variables. In addition, considering the effect of the exercise mode on each indicator [32,33], we grouped the data into two exercise modes, 'cycling' or 'running', and calculated the mean and 95% CI for each indicator for each exercise mode.

## Results

### Accepted article and data extraction for the Japanese population

We obtained 62 candidate articles through a systematic search using PubMed and Ichushi-Web. Seventeen articles were excluded in the primary screening, 38 articles were excluded in the secondary screening (articles with missing required data = 20, articles comprising unhealthy individuals = 7, articles without full text = 4, articles not comprising Japanese individuals = 1, articles with duplication = 6), and 7 articles were accepted [34–40]. The articles on aerobic capacity in the Japanese population were old, written in Japanese, and unavailable in the database. Thus, 14 articles were added through a grey literature search using Google Scholar [29,41–53]. Consequently, 21 articles [29,34–53] were included in the meta-analysis (Fig 1). Accepted articles and data extraction for other populations are shown in the Supplemental Data.

**PRISMA 2020 flow diagram for new systematic reviews which included searches of databases, registers and other sources**

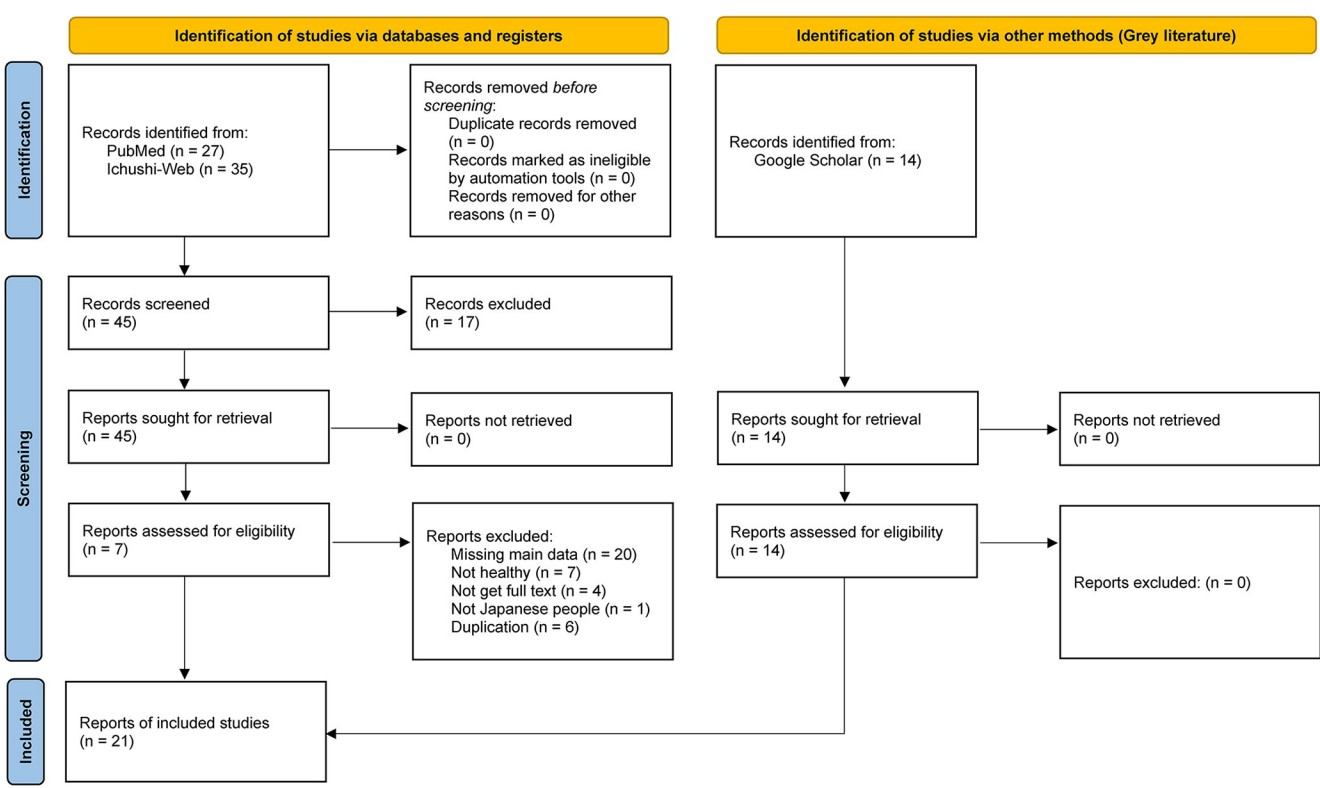

*From:* Page MJ, McKenzie JE, Bossuyt PM, Boutron I, Hoffmann TC, Mulrow CD, et al. The PRISMA 2020 statement: an updated guideline for reporting systematic reviews. BMJ 2021;372:n71. doi: 10.1136/bmj.n71.

**Fig 1. Flow diagram of the article search process.** Sixty-two articles were identified in the systematic search, 45 were selected in the evaluation by title and abstract (primary screening), and 7 were included in the evaluation by full-text close reading (secondary screening). Furthermore, the grey literature search was performed on the reference lists of the articles identified in the systematic search, finding additional 14 articles. Therefore, 21 articles were finally included and combined in meta-analysis.

The systematic search extracted descriptive statistics data of $\dot{V}O_2peak/kg$ in Japanese from 22 studies on men (13 cycle, 59.1%; 9 run, 40.9%) and 17 studies on women (11 cycle, 64.7%; 6 run, 35.3%) and analysed 78,714 participants (54,614 men, 69.4%; 24,100 women, 30.6%) (S3 and S4 Tables in S1 File).

The systematic search extracted descriptive statistics data of AT/kg in Japanese from 9 studies on men (7 cycle, 77.8%; 2 run, 22.2%) and 8 studies on women (6 cycle, 75.0%; 2 run, 25.0%) and analysed 4,042 participants (1,961 men, 48.5%; 2,081 women, 51.5%) (S7 and S8 Tables in S1 File).

## Relationship between the estimated standard values of $\dot{V}O_2peak/kg$ and age for Japanese population

Tables 1 and 2 show the estimated standard values of $\dot{V}O_2peak/kg$ by sex and age in Japanese population. S11 and S12 Tables in S1 File also show data for other populations. In addition, the relationship between the estimated standard value of $\dot{V}O_2peak/kg$ (cycle and run combined) and age is shown in S2 Fig in S1 File. $\dot{V}O_2peak/kg$ was the highest during the ages of 10–19 years in Japanese population regardless of sex and progressively declined with ageing after age 20–29 years (S2C and S2F Fig in S1 File, white circles). This study showed that $\dot{V}O_2peak/kg$ declined by –7.0% for Japanese men and –6.5% for Japanese women for each

**Table 1. Estimated standard values of weight-adjusted peak oxygen uptake ($\dot{V}O_2peak/kg$, mL/kg/min) and anaerobic threshold (AT/kg, mL/kg/min) for men in each age group of the Japanese population.**

| Men | All | | Cycle | | Run | |
|---|---|---|---|---|---|---|
| Age | Mean | 95% CI | Mean | 95% CI | Mean | 95% CI |
| (years) | (mL/kg/min) | | (mL/kg/min) | | (mL/kg/min) | |
| $\dot{V}O_2peak/kg$ | | | | | | |
| 4–9 | 47.6 | [45.5, 49.7] | 49.0 | [46.1, 51.9] | 46.9 | [44.4, 49.4] |
| 10–19 | 51.2 | [45.4, 57.0] | 52.2 | [44.2, 60.2] | 49.3 | [42.4, 56.2] |
| 20–29 | 43.2 | [39.8, 46.6] | 41.4 | [37.8, 45.0] | 47.0 | [41.4, 52.6] |
| 30–39 | 37.2 | [34.3, 40.1] | 35.9 | [32.7, 39.1] | 39.1 | [34.1, 44.1] |
| 40–49 | 34.5 | [32.1, 36.9] | 33.5 | [30.5, 36.5] | 36.0 | [32.1, 39.9] |
| 50–59 | 31.7 | [29.3, 34.1] | 31.0 | [27.9, 34.1] | 33.0 | [29.7, 36.3] |
| 60–69 | 28.6 | [26.3, 30.9] | 27.1 | [25.3, 28.9] | 30.6 | [26.6, 34.6] |
| 70–79 | 26.3 | [24.8, 27.8] | 25.0 | [24.5, 25.5] | 27.3 | [25.9, 28.7] |
| 80–89 | N/A | N/A | N/A | N/A | N/A | N/A |
| AT/kg | | | | | | |
| 4–9 | N/A | N/A | N/A | N/A | N/A | N/A |
| 10–19 | N/A | N/A | N/A | N/A | N/A | N/A |
| 20–29 | 21.1 | [18.3, 23.9] | 19.4 | [18.9, 19.9] | 24.6 | [17.7, 31.5] |
| 30–39 | 18.3 | [16.7, 19.9] | 17.2 | [16.4, 18.0] | 20.9 | [18.1, 23.7] |
| 40–49 | 16.8 | [15.3, 18.3] | 15.8 | [15.0, 16.6] | 20.3 | [17.1, 23.5] |
| 50–59 | 15.9 | [14.5, 17.3] | 14.9 | [14.2, 15.6] | 19.4 | [18.5, 20.3] |
| 60–69 | 15.8 | [14.0, 17.6] | 14.6 | [13.5, 15.7] | 20.2 | [18.1, 22.3] |
| 70–79 | 15.2 | [11.4, 19.0] | 13.5 | [10.6, 16.4] | 18.7 | N/A |
| 80–89 | 10.9 | N/A | 10.9 | N/A | N/A | N/A |

$\dot{V}O_2peak/kg$, weight-adjusted peak oxygen uptake; **AT/kg**, weight-adjusted anaerobic threshold; **95% CI**, 95% confidence interval; **N/A**, data not available for meta-analysis. The '/kg' here means weight correction, not lean mass.

**Table 2. Estimated standard values of weight-adjusted peak oxygen uptake ($\dot{V}O_2$peak, mL/kg/min) and anaerobic threshold (AT/kg, mL/kg/min) for women in each age group of the Japanese population.**

| Women | All | | Cycle | | Run | |
|---|---|---|---|---|---|---|
| Age | Mean | 95% CI | Mean | 95% CI | Mean | 95% CI |
| (years) | (mL/kg/min) | | (mL/kg/min) | | (mL/kg/min) | |
| $\dot{V}O_2$peak/kg | | | | | | |
| 4–9 | 42.0 | [38.1, 45.9] | 43.0 | [42.3, 43.7] | 41.3 | [34.3, 48.3] |
| 10–19 | 43.2 | [37.2, 49.2] | 45.0 | [36.3, 53.7] | 40.4 | [31.0, 49.8] |
| 20–29 | 33.6 | [30.3, 36.9] | 32.1 | [28.2, 36.0] | 36.5 | [30.7, 42.3] |
| 30–39 | 30.6 | [28.2, 33.0] | 28.9 | [27.3, 30.5] | 34.0 | [28.8, 39.2] |
| 40–49 | 27.4 | [25.5, 29.3] | 26.0 | [24.6, 27.4] | 30.6 | [26.4, 34.8] |
| 50–59 | 25.6 | [24.0, 27.2] | 24.4 | [23.1, 25.7] | 28.2 | [24.9, 31.5] |
| 60–69 | 23.4 | [21.4, 25.4] | 21.7 | [20.5, 22.9] | 27.9 | [25.2, 30.6] |
| 70–79 | 23.1 | [20.4, 25.8] | 21.2 | [20.5, 21.9] | 25.1 | [21.7, 28.5] |
| 80–89 | N/A | N/A | N/A | N/A | N/A | N/A |
| AT/kg | | | | | | |
| 4–9 | N/A | N/A | N/A | N/A | N/A | N/A |
| 10–19 | N/A | N/A | N/A | N/A | N/A | N/A |
| 20–29 | 17.4 | [16.0, 18.8] | 16.4 | [15.8, 17.0] | 19.4 | [17.3, 21.5] |
| 30–39 | 17.0 | [15.7, 18.3] | 16.2 | [15.3, 17.1] | 19.1 | [18.3, 19.9] |
| 40–49 | 15.7 | [14.2, 17.2] | 14.7 | [13.6, 15.8] | 18.6 | [16.7, 20.5] |
| 50–59 | 15.0 | [13.4, 16.6] | 14.0 | [12.8, 15.2] | 17.8 | [15.4, 20.2] |
| 60–69 | 14.5 | [13.1, 15.9] | 13.6 | [12.4, 14.8] | 17.2 | [16.3, 18.1] |
| 70–79 | 14.2 | [9.7, 18.7] | 12.1 | [9.4, 14.8] | 18.5 | N/A |
| 80–89 | 10.6 | N/A | 10.6 | N/A | N/A | N/A |

$\dot{V}O_2$peak/kg, weight-adjusted peak oxygen uptake; **AT/kg**, weight-adjusted anaerobic threshold; **95% CI**, 95% confidence interval; **N/A**, data not available for meta-analysis. The '/kg' here means weight correction, not lean mass.

decade after the age 20–29 years. In both Japanese men and women, the degree of decline in $\dot{V}O_2$peak/kg was greater during the ages of 20–29 years than during the ages of > 30–39 years (S2C and S2F Fig in S1 File, white circles). To calculate the rate of increase in $\dot{V}O_2$peak/kg with growth during the ages of 10–19 years and the rate of decline in $\dot{V}O_2$peak/kg with ageing after the ages of 20–29 years, linear regression equations with age and $\dot{V}O_2$peak/kg as variables were used (Table 3).

In the comparison of the estimated standard values of $\dot{V}O_2$peak/kg by exercise mode, the values for Japanese men (mean, –1.8 mL/kg/min [–5.3%]; range, 2.9 to –5.6 [5.9 to –11.8%] mL/kg/min) and women (mean, –2.7 mL/kg/min [–9.7%]; range, 4.6 to –6.2 [11.4 to –22.2%] mL/kg/min) were lower for cycling than for running (Tables 1 and 2).

## Relationship between the estimated standard values of AT/kg and age for the Japanese population

Tables 1 and 2 show the estimated standard values of AT/kg by sex and age in the Japanese population. In addition, the relationship between the estimated standard value of AT/kg (cycle and run combined) and age is shown in S3 Fig in S1 File. In the Japanese population, AT/kg was highest in both men and women aged 20–29 years and progressively declined with age (S3C and S3F Fig in S1 File). This study showed that AT/kg declined by –5.3% in Japanese

**Table 3. Equations for estimating the standard values of weight-adjusted peak oxygen uptake ($\dot{V}O_2$peak/kg, mL/kg/min) and anaerobic threshold (AT/kg, mL/kg/min) in each age group of the Japanese population.**

| Japanese population | | Age | All | | Bicycle | | Run | |
|---|---|---|---|---|---|---|---|---|
| | | | Equation | $R^2$ | Equation | $R^2$ | Equation | $R^2$ |
| $\dot{V}O_2$peak/kg | | | | | | | | |
| Men | Japan | $\leq 19$ | Y = 0.03x + 48.3 | 0.00 | Y = -0.11x + 49.9 | 0.00 | Y = 0.12x + 47.5 | 0.01 |
| | Japan | $\geq 20$ | Y = -0.36x + 51.6 | 0.50 | Y = -0.34x + 49.3 | 0.56 | Y = -0.41x + 56.4 | 0.55 |
| Women | Japan | $\leq 19$ | Y = -0.46x + 46.9 | 0.14 | Y = -0.67x + 50.0 | 0.23 | Y = -0.42x + 46.2 | 0.12 |
| | Japan | $\geq 20$ | Y = -0.26x + 40.3 | 0.46 | Y = -0.25x + 38.4 | 0.63 | Y = -0.27x + 44.5 | 0.51 |
| AT/kg | | | | | | | | |
| Men | Japan | $\leq 19$ | N/A | N/A | N/A | N/A | N/A | N/A |
| | Japan | $\geq 20$ | Y = -0.12x + 22.8 | 0.34 | Y = -0.11x + 21.3 | 0.71 | Y = -0.10x + 25.4 | 0.35 |
| Women | Japan | $\leq 19$ | N/A | N/A | N/A | N/A | N/A | N/A |
| | Japan | $\geq 20$ | Y = -0.08x + 19.5 | 0.29 | Y = -0.09x + 18.8 | 0.57 | Y = -0.04x + 20.3 | 0.31 |

$\dot{V}O_2$**peak/kg**, weight-adjusted peak oxygen uptake; **AT/kg**, weight-adjusted anaerobic threshold; $R^2$, coefficient of determination; **N/A**, data not available for meta-analysis. Y is $\dot{V}O_2$peak/kg or AT/kg, X is Age. The '/kg' here means weight correction, not lean mass.

men and –4.1% in Japanese women in each decade after 20–29 years of age. The linear regression equation between age and AT/kg is shown in Table 3.

In the comparison of the estimated standard values of AT/kg by exercise mode, the values for Japanese men (mean, –4.8 mL/kg/min [–23.3%]; range, –3.7 to –5.6 [–17.7 to –27.8%] mL/kg/min) and women (mean, –3.9 mL/kg/min [–21.4%]; range, –2.9 to –6.4 [–15.2 to –34.6%] mL/kg/min) were lower for cycling than for running (Tables 1 and 2).

## Estimated standard values of $\dot{V}O_2$peak/kg and AT/kg for the other populations

As a second aim of this study, an umbrella review was conducted to estimate standard values of aerobic capacity in other populations. Please find supplementary materials (S5, S6 and S9-S13 Table in S1 File). The results are briefly summarised as follows. A total of 36 articles were included in the meta-analysis (S1 Fig in S1 File, combined with cycle and run). S11 and S12 Tables in S1 File show the estimated standard values of $\dot{V}O_2$peak/kg and AT/kg by sex and age in the other populations. In the comparison of the estimated standard values of $\dot{V}O_2$peak/kg between Japanese and other populations, the values for Japanese men (–2.6 mL/kg/min, –6.6%) and women (–1.7 mL/kg/min, –5.4%) were lower than those for the other populations (S2C and S2F Fig in S1 File). In the comparison of the estimated standard values of AT/kg between Japanese and other populations, Japanese men (–4.1 mL/kg/min, –19.5%) and women (–2.1 mL/kg/min, –11.9%) had lower values than the other populations (S3C and S2F Fig in S1 File).

## Discussion

### Main findings

The main findings of this study are as follows. First, the estimated standard values of $\dot{V}O_2$peak/kg and AT/kg declined with age in the Japanese population after age 20–29 years. Second, $\dot{V}O_2$peak/kg tended to be lower in Japanese men (–2.6 mL/kg/min, –6.6%) and women (–1.7 mL/kg/min, –5.4%) than in individuals from other populations. Third, AT/kg

tended to be lower in Japanese men (–4.1 mL/kg/min, –19.5%) and women (–2.1 mL/kg/min, –11.9%) than in individuals from other populations. To the best of our knowledge, this is the first study to estimate the standard values for $\dot{V}O_2peak/kg$ and AT/kg by sex and age from previous studies for a large number of Japanese individuals and to compare the data with those from other populations.

## Relationship between the estimated standard values of $\dot{V}O_2peak/kg$ and age

This study showed that $\dot{V}O_2peak/kg$ declined for Japanese men and women with ageing after the age of 20 years. However, the rate of decline in $\dot{V}O_2peak/kg$ per decade was not uniform across all ages, with a relatively large drop between the ages of 15–29 years, followed by a relatively small decline after age 40 years (S2C and S2F Fig in S1 File, white circles). This result is consistent with the findings of Hawkins and Wiswell [54], who suggested that the rate of decline in $\dot{V}O_2peak/kg$ varied across age groups. It was assumed that the sharp decline in $\dot{V}O_2peak/kg$ during adolescence was caused by the interaction of two factors. One is the effect of the increase in body weight and fat resulting from growth and sexual characteristics during the 15–19 years and from the lifestyle changes associated with employment during age 20–29 years. The other is the effects of the decline in absolute $\dot{V}O_2peak$ with ageing, which is discussed below.

Age-related decline in $\dot{V}O_2peak$ is mainly caused by declines in maximal cardiac output (central factor) and lean body mass (peripheral factor) [54]. Age-related decline in maximal cardiac output is mainly due to a decline in peak heart rate [55]. The peak heart rate declined linearly at a rate of 7 beats/min per decade according to Tanaka et al.'s estimated equation ($Y = 208 – 0.7 \times age$) [56]; the resting heart rate was constant (approximately 70 beats/min) [57]. From these data, the heart rate reserve (peak heart rate–resting heart rate) was calculated, and the average rate of decline in heart rate reserve per decade was estimated to be –6.2%. Therefore, the decline in $\dot{V}O_2peak$ was mostly attributable to a decline in peak heart rate. One of the peripheral factors, muscle mass (lean body mass), especially lower limb muscle mass, has been reported to decline in Japanese men (30.9%) and women (28.5%) between the ages of 20 and 80 years (average decline of approximately 4% per decade) [58]. This decline in lower limb muscle mass, together with the above-mentioned decline in peak heart rate, a central factor, is thought to contribute to the age-related decline in $\dot{V}O_2peak$.

In the comparison of the estimated standard values of $\dot{V}O_2peak/kg$ between Japanese and other populations, the values for Japanese men and women were lower than those of other populations. Shvartz and Reibold (1990) also reported that $\dot{V}O_2peak/kg$ was lower in Japanese men (3%) and women (5%) than in individuals of other populations [23]. Considering that $\dot{V}O_2peak/kg$ is more strongly related to lean body mass than to body weight [59], the difference in $\dot{V}O_2peak/kg$ may have been influenced by the lower lean body mass and/or metabolic rate of the younger generation of Japanese individuals [58]. In addition, the World Health Organization has reported that the prevalence of physical inactivity are 27.5% worldwide and 35.5% in Japan (95% CI: 20.5–53.8%) [60]. It is speculated that this may also have led to the slightly lower value of the $\dot{V}O_2peak/kg$ in Japanese rather than other populations.

## Relationship between the estimated standard values of AT/kg and age

AT is one of the indicators observed to be submaximal in incremental exercise tests, in contrast to $\dot{V}O_2peak$, which requires maximal effort until exhaustion. Because AT can be measured

and evaluated more safely and with less burden on the participant than $\dot{V}O_2$peak, it has been used in medical practice as a powerful predictor of mortality risk, especially in postoperative patients [61], organ transplant patients [62,63], and chronic heart failure patients [19]. However, available data on the standard values of AT in the general population are lacking.

This study showed that AT/kg declined in Japanese men (−5.3%) and women (−4.1%) each decade after 20–29 years of age. However, this age-related decline in AT/kg was not uniform and showed a different trend from the decline in $\dot{V}O_2$peak/kg (S3C and S3F Fig in S1 File, white circles). In both men and women, the rate of decline was linear from the ages 20–29 to 40–49 years and relatively slowed down subsequently. It is difficult to explain this observation, because the amount of AT/kg data was small (only approximately 4,000 participants), particularly in the 70–89 year group. Therefore, there is a need to accumulate further data on AT/kg to improve the validity of the estimated standard values of AT/kg.

The percentage of AT to $\dot{V}O_2$peak (AT/$\dot{V}O_2$peak) provides a physiological interpretation when assessing endurance characteristics [64] and is usually approximately 50% in healthy individuals [4,65,66]. In this study, the AT/$\dot{V}O_2$peak was calculated based on the estimated data of $\dot{V}O_2$peak/kg and AT/kg, although the data were not analysed and compared in the same individuals. The AT/$\dot{V}O_2$peak increased with ageing in this study (AT/$\dot{V}O_2$peak was 48.8, 49.2, 48.7, 50.2, 55.2 and 57.8% in men, and 51.8, 55.6, 57.3, 58.6, 62.0 and 61.5% in women, in the age groups 20–29, 30–39, 40–49, 50–59, 60–69 and 70–79 years, respectively). These results are consistent with some previous studies [67–69]. The reasons for this phenomenon may be explained by the facts that 1) the underlying physiological mechanisms of $\dot{V}O_2$peak and AT are different, therefore, the effects of ageing differ between them, and 2) AT has greater trainability than $\dot{V}O_2$peak in middle-aged and older individuals.

Previous studies suggest that maintaining the physiologically determinant factor of $\dot{V}O_2$peak as individuals age is challenging, whereas maintaining the physiologically determinant factor of AT is more achievable [69,70]. $\dot{V}O_2$peak depends on the ability of the respiratory and circulatory systems to transport oxygen and the ability of the skeletal muscle to utilise oxygen, whereas AT largely depends on the oxidative capacity of the skeletal muscle [64]. Several studies showed that respiratory and circulatory capacity steadily declined with ageing [71–76], whereas the oxidative capacity of the skeletal muscle was relatively maintained with ageing [77–84]. In summary, the effect of ageing was smaller for AT/kg (mainly dependent on peripheral factors) than for $\dot{V}O_2$peak/kg (dependent on central and peripheral factors), suggesting that AT/$\dot{V}O_2$peak increases with ageing.

Notably, beyond the differences in $\dot{V}O_2$peak (S2 Fig in S1 File), AT in Japanese men aged 20–69 years was markedly lower than that in other populations (S3 Fig in S1 File). The mechanisms contributing to this phenomenon are unknown. However, given that AT largely depends on the skeletal muscle's oxidative capacity, the skeletal muscle mass or skeletal muscle's oxidative capacity may differ between Japanese and other populations. Silva et al. (2010) reported that skeletal muscle mass decreases after 27 years of age, and that ethnic differences exist in this ageing phenomenon of skeletal muscle mass. They also reported that skeletal muscle mass is lowest in Asians, including Japanese; however, only women were included in their study [85]. Further research should be conducted on the mechanisms underlying AT differences between ethnic groups with data from both sexes.

### Differences in exercise mode

Comparing the estimated standard values of $\dot{V}O_2$peak/kg and AT/kg by exercise mode, the Japanese population had lower $\dot{V}O_2$peak/kg and AT/kg values during cycling than running.

Compared with running, cycling $\dot{V}O_2$peak/kg was lower for men (–5.3%) and women (–9.7%). These results are consistent with the findings of previous studies that reported a 5%–22% lower $\dot{V}O_2$peak/kg for cycling than running [86–91]. In contrast, $\dot{V}O_2$peak/kg was lower for running than for cycling in Japanese men and women aged 4–9 and 10–19 years. Although the detailed reasons for this observation are beyond the scope of this review, it may be due to sampling bias rather than the physiological effects of the different exercises. Additionally, the reversal phenomenon of cycling and running may presumably be due to the small number of studies on the aerobic capacity of minors, and the inclusion of sports children in the studies using cycling (Tamiya, 1991) [34].

Compared with running, cycling AT/kg was lower for men (–23.3%) and women (–21.4%) in the Japanese population. Moreover, AT/kg was –11% to –16% lower in triathletes in cycling than in running [92]. The present study did not compare the effects of exercise mode on $\dot{V}O_2$peak/kg and AT/kg in the same individuals, which may indicate inter-individual differences.

## Strengths and limitations

The strengths of this study are as follows. Based on large-scale data ($\dot{V}O_2$peak/kg for 78,714 and AT/kg for 4,042) collected from several studies, it was possible to evaluate the aerobic capacity of Japanese populations according to sex and age. The present data may be more representative of the population results because the mean and distribution of aerobic capacity were estimated from multiple surveys. The estimated standard value of aerobic capacity obtained in this study is expected to be utilised in research and education in the fields of public health, physical education, and sports.

A limitation of this study is that some biases cannot be completely excluded, such as the effect on values due to differences in research methods among the collected articles. First, it appears likely that the data collected in this study were obtained only from individuals with a high physical fitness level and high motivation to participate in physical fitness tests, regardless of their population. Second, the comparison of estimated standard values by age in this study was based on a cross-sectional analysis that may be affected by confounders and cohort effects and thus may not detect true age-related changes in aerobic capacity independent of these effects [93]. Finally, the estimated standard values of aerobic capacity of Japanese individuals in this study are based on data obtained during the past 40 years, between 1977 and 2013, which may be different from the current status of aerobic capacity of the Japanese population. Therefore, the estimated standard values of $\dot{V}O_2$peak/kg and AT/kg obtained in this study should be improved and validated based on further data accumulation in future studies.

## Perspectives

Identifying the standard values of aerobic capacity within a population serves a twofold purpose: aiding in the evaluation of individuals' health statuses and the formulation of effective health promotion strategies. Assessment of these factors requires population information, especially distribution. Furthermore, when devising reference values to guide health promotion initiatives, it becomes imperative to strike a balance between specific benefits and the rate at which the population adheres to these guidelines. Therefore, the standard values estimated from this study may contribute to assessing and developing health promotion strategies based on a more holistic understanding of the population, in addition to determining the aerobic capacity and health status of the Japanese population.

The literature research method used in this study can be used in Japan and in other populations worldwide. This research method can contribute to understanding the distribution of

aerobic capacity in one's own population, which is difficult to practice in a single study. As a result, it may be able to significantly contribute to understanding the health status of its own population.

Evaluation of aerobic capacity requires an exercise test in the laboratory, for which breath gas analysis is the standard method. Recently, attention has been focused on the development of wearable devices that estimate aerobic capacity based on data, such as pulse rate, acceleration, and distance travelled [94,95]. Therefore, it is necessary to improve the accuracy of these estimation methods [96,97] and accumulate further evidence to determine the standard values of aerobic capacity based on these estimation methods.

There is a need to update the estimated standard values based on the accumulation of the latest research data, because aerobic capacity may potentially change (unfortunately, probably decline) with long-term changes in lifestyle associated with the progress in science and technology.

## Conclusions

This study estimated the standard values and distribution for aerobic capacity ($\dot{V}O_2$peak/kg and AT/kg) in the current Japanese population based on meta-analysis. These standard values may help to set and update the reference values for health promotion, because the reference values should reflect not only the epidemiological evidence but the physiological background in the target population and feasibility. The estimation of standard values by meta-analysis attempted in this paper can be conducted in any country or region and be used for public health purposes.

## Supporting information

**S1 Checklist. Preferred Reporting Items for Systematic reviews and Meta-Analyses extension for Scoping Reviews (PRISMA-ScR) checklist.**
(DOCX)

**S1 File. S1-S13 Table, S1-S3 Fig., Supplemental Methods, and Supplemental Results.**
(DOCX)

**S2 File. Evidence table for studies of Japanese population.**
(XLSX)

**S3 File. Evidence table for studies of other population.**
(XLSX)

## Acknowledgments

We would like to express our appreciation to the authors of the studies included in this study and to all individuals involved in the data collection. We would like to thank Editage (www. editage.jp) for English language editing.

## Author Contributions

**Conceptualization:** Hiroshi Akiyama, Daiki Watanabe, Motohiko Miyachi.

**Data curation:** Hiroshi Akiyama, Motohiko Miyachi.

**Formal analysis:** Hiroshi Akiyama, Daiki Watanabe, Motohiko Miyachi.

**Funding acquisition:** Motohiko Miyachi.

**Investigation:** Hiroshi Akiyama, Daiki Watanabe, Motohiko Miyachi.

**Methodology:** Hiroshi Akiyama, Daiki Watanabe, Motohiko Miyachi.

**Project administration:** Hiroshi Akiyama, Motohiko Miyachi.

**Supervision:** Motohiko Miyachi.

**Validation:** Motohiko Miyachi.

**Visualization:** Hiroshi Akiyama, Motohiko Miyachi.

**Writing – original draft:** Hiroshi Akiyama, Motohiko Miyachi.

**Writing – review & editing:** Hiroshi Akiyama, Daiki Watanabe, Motohiko Miyachi.

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
