## [Decision Letter · Decision Letter 0]

13 Jul 2023

PONE-D-23-15750Estimated standard values of aerobic capacity according to sex and age in a Japanese population: a scoping reviewPLOS ONE

Dear Dr. Miyachi,

Thank you for submitting your manuscript to PLOS ONE. After careful consideration, we feel that it has merit but does not fully meet PLOS ONE’s publication criteria as it currently stands. Therefore, we invite you to submit a revised version of the manuscript that addresses the points raised during the review process.

We look forward to receiving your revised manuscript.

Kind regards,

Yosuke Yamada

Academic Editor

PLOS ONE

Journal Requirements:

"YES-This work was supported by the National Research and Development Agency (JST), SPRING (Grant Number JPMJSP2128 to Hiroshi Akiyama) and Practical Research Project for Lifestyle-related Diseases including Cardiovascular Diseases and Diabetes Mellitus from the Ministry of Health, Labour and Welfare (Grant Number 21FA1004 and 22FA1004 to Motohiko Miyachi)."

4. Please expand the acronym “JST” (as indicated in your financial disclosure) so that it states the name of your funders in full.

Reviewers' comments:

Reviewer's Responses to Questions

**Comments to the Author**

1. Is the manuscript technically sound, and do the data support the conclusions?

Reviewer #1: Partly

Reviewer #2: Yes

2. Has the statistical analysis been performed appropriately and rigorously? 

Reviewer #1: Yes

Reviewer #2: Yes

3. Have the authors made all data underlying the findings in their manuscript fully available?

Reviewer #1: Yes

Reviewer #2: Yes

4. Is the manuscript presented in an intelligible fashion and written in standard English?

Reviewer #1: No

Reviewer #2: Yes

5. Review Comments to the Author

Reviewer #1: General comments

The authors performed a meta-analysis to estimate the standard values of VO2peak and AT in the Japanese and other populations stratified by sex and age. The theme of the present study is potentially important. Overall, the statistical analyses and data reporting are appropriate. However, unfortunately the manuscript is not well written. This reviewer has a number of comments to improve the quality of the manuscript.

Specific comments

Page 2, line 29: The words “VO2peak/kg and AT/kg in other populations” suddenly appeared in the latter part of Abstract. The authors should explain beforehand on the aerobic capacity in the other population in the background, purpose, and methods sections.

Page 2, lines 31-32: Readers may misunderstand the conclusion sentence. Some readers may think that the decline in the aerobic capacity is lower (or smaller) in Japanese population than the other population when they read only abstract. Please revise the sentence to prevent the potential readers’ confusion.

Page 3, lines 50-55: To the reviewer’s understanding, epidemiological evidence regarding the association between aerobic capacity and disease risk or mortality risk is absolutely necessary to establish a reference value of aerobic capacity. This reviewer cannot understand the reason why the standard value of the aerobic capacity is absolutely needed to establish the reference value of aerobic capacity. Please explain the reason more clearly and concretely.

Page 4, lines 60-68: The reviewer could not clearly understand the rationales of the primary and secondary aims of the present study. It is even unclear that the authors are trying to explain on the primary or secondary aims in the first paragraph of page 4. The authors should clearly mention the rationales of the primary and secondary aims in this paragraph. The authors should focus on mentioning only the purpose of the present study in the last paragraph of the Introduction section.

Page 4, line 71: General readers who are not familiar with meta-analysis may have no idea about the terms “scoping review” as well as “grey literature search”, or “umbrella review”. Please explain more courteously on them in the Materials and methods section or other appropriate sections.

Page 4, lines: 60-61: The sentence is poorly written, and needs to be revised.

Page 5, lines 94-101: VO2max in the selected articles is directly measured by expired gas during maximal exercise test, or it is indirectly estimated by heart rate response during submaximal exercise test (e.g., using an equation of estimated maximum heart rate 220-age)? It is not clearly written whether the directly or indirectly measured VO2max or both are included (as VO2peak) in the present meta-analysis.

Page 6, 102-110: Are these criteria are the criteria for Japanese population or other population or both? The authors performed a meta-analysis of other populations than Japanese population as a secondary aim (one of the major aims of the present study), but they mentioned “studies not in Japanese” as a criterion (line 106).

Please be explicit not only about the eligibility criteria, but also the search strategy, study selection, data analysis for each of the Japanese population analysis and other population analysis.

Page 6, lines 109-110: The sentence is poorly written, and needs to be revised.

Page 6, lines 113-115: There is no description on the roles of the primary and secondary screening. There is also no explanation on the required conditions that articles pass the primary screening to the secondary screening. Please be explicit about that these primary and secondary screenings are used for Japanese population, other population, or both.

In addition, for example, the authors mentioned that “The following items were extracted: (1) first author’s name, (2) year of publication, (3) sex, (4) age, (5) exercise mode, and (6) means and distributions of VO2peak and AT (SD, standard error, and confidence interval [CI])”. Please be explicit about that this (and other each procedure) is conducted as the primary screening or secondary screening.

Page 7, line 127: VO2peak (or VO2max) is sometimes expressed as / kg lean body mass (LBM). The authors should clearly mention at least once that they used /kg body weight (not LBM) in the Materials and methods section (not only in footnotes of Tables and legends of Figures but also in the Methods section).

Page 7, lines 130-131: The reviewer could not understand the meaning of the sentence. Do you mean that the article (ref. 39) reported VO2peak and AT measured by indirect method? It is known that the VO2peak and AT measured by indirectly method are underestimated when they are compared with VO2peak and AT measured by directly method?

Page 7, lines 132-134: The authors showed VO2peak and AT data by 10-year age group in Tables 1 and 2. However, they suddenly explain on the categorization of ≤19 years or ≥20 years, before they explain about the 10-year age group categorization first. It is also unclear that these data analyses are applied to Japanese population or other population, or both populations. Data analysis section is poorly written and should be drastically revised to prevent readers’ confusion.

Page 7, lines 135-136: The authors mentioned “The slopes and intercepts of the mean and age for each indicator from each study were determined using…”. Is it correct? It should be “The slopes and intercepts of the association of age with VO2peak and AT for each age category (≦19 years, ≧20 years) were determined using…”.

Page 8, line 142: Please be explicit about that these are accepted articles and data extraction for Japanese population, or other population than Japanese, or both.

Page 10, Table 1: The authors may touch the possible reason why the VO2peak data measured by run are lower than VO2peak data measured by cycle in the age groups of 4-9 and 10-19 in the Materials and methods section or Discussion section.

Page 10, Table 1: The percentage of AT in VO2peak is approximately 50% in the age groups of young and middle-aged men (Table 1) and women (Table 2), but the percentage is apparently higher in older men and women (55-61% in Tables 1 and 2). To the reviewer’s knowledge, endurance athletes have a higher percentage of AT in VO2peak (>60%), but not in older people. How do the authors explain this phenomenon?

Page 13, lines 216-220: It is unclear that the VO2peak and AT data shown in S2 and S3 figures are measured by cycle, run, or combined with cycle and run.

Page 16, lines 273-279: Based on the AT data in the other age groups than 70–89-year group, it seems that the decrease in AT in Japanese population compared with other population (shown in S3 figure) is greater than the decrease in VO2peak in the Japanese population compared with other population (shown in S2 figure). How do the authors explain this interesting phenomenon?

Page 18, line 310: The authors stated that “identifying the distribution of aerobic capacity in population may help estimate their health status and develop a health promotion strategy” in Abstract. In addition, they mentioned that “estimated standard values may help set or update more practical reference values for health promotion in the Japanese population” in the conclusion. However, this reviewer could not clearly understand the reason why the estimated standard values may help set or update more practical reference values for health promotion in the Japanese population. Please describe more how the standard values of VO2peak and AT estimated by the present study actually contribute to the development of the reference value of the aerobic capacities for future health promotion, in this Perspectives section or other appropriate sections, more clearly and concretely.

Reviewer #2: Line 281.

VO2peak/kg is lower in Run than in Cycle for both males and females in their teens.

Perhaps this is a discrepancy arising from Tamiya's (1991) higher results for teenagers with Cycle.

The author should add a discussion of this result.

(No need to re-review.)

6. PLOS authors have the option to publish the peer review history of their article (what does this mean?). If published, this will include your full peer review and any attached files.

Reviewer #1: No

Reviewer #2: **Yes: **Kojiro Ishii

---

## [Author Response · Author response to Decision Letter 0]

15 Aug 2023

We would like to thank the reviewers for reading the manuscript and providing their constructive comments. Our responses to the reviewers are listed in "Response to Reviewers" file and the corresponding corrections have been made in the "Revised Manuscript with Track Changes". The reviewers' comments (in black) are listed below, followed by our responses (in red).

Responses to Reviewer 1’s comments

Comment #1 (General comments): The authors performed a meta-analysis to estimate the standard values of VO2peak and AT in the Japanese and other populations stratified by sex and age. The theme of the present study is potentially important. Overall, the statistical analyses and data reporting are appropriate. However, unfortunately the manuscript is not well written. This reviewer has a number of comments to improve the quality of the manuscript.

Our response #1: We appreciate the constructive comments of the reviewer and have revised the manuscript accordingly. In addition, the revised manuscript was proofread by an English proofreading company. We hope these measures have helped improve the manuscript and enhance the reader's understanding.

Comment #2: Page 2, line 29: The words “VO2peak/kg and AT/kg in other populations” suddenly appeared in the latter part of Abstract. The authors should explain beforehand on the aerobic capacity in the other population in the background, purpose, and methods sections.

Our response #2: As per the reviewer’s suggestion, we have described the inclusion of aerobic capacity data from other populations in the background, purpose, and methods subsections of the Abstract. However, this information has been only briefly provided to avoid confusion among readers regarding the primary purpose of this meta-analysis, which was to obtain a standard value estimated of aerobic capacity in the Japanese population (Page 2, Lines 20, 23-24).

Comment #3: Page 2, lines 31-32: Readers may misunderstand the conclusion sentence. Some readers may think that the decline in the aerobic capacity is lower (or smaller) in Japanese population than the other population when they read only abstract. Please revise the sentence to prevent the potential readers’ confusion.

Our response #3: We have revised the sentence to avoid confusion among the readers (Page 2, Lines 31-32).

Comment #4: Page 3, lines 50-55: To the reviewer’s understanding, epidemiological evidence regarding the association between aerobic capacity and disease risk or mortality risk is absolutely necessary to establish a reference value of aerobic capacity. This reviewer cannot understand the reason why the standard value of the aerobic capacity is absolutely needed to establish the reference value of aerobic capacity. Please explain the reason more clearly and concretely.

Our response #4: We recognize that the determination of reference values must be based on epidemiological evidence regarding the association between aerobic capacity and risk of disease and mortality, as well as the physiological background of the target population and the feasibility. In other words, the target values in the guidelines need to be set in consideration of both the ideal (reference value) and the reality (standard value). For example, in the Dietary Reference Intakes for Japanese 2020, a value of adequate intake for sodium (salt equivalent) was set considering the epidemiological evidence, as well as recent trends in salt intake of the target population and the feasibility. Therefore, standard values representing the real-world scenario are needed to set reference values that can be achieved by a greater number of individuals in the target population. These reasons have been stated in the third paragraph of the Introduction (Page 3, Line 51 ~ Page 4, Line 59).

Comment #5: Page 4, lines 60-68: The reviewer could not clearly understand the rationales of the primary and secondary aims of the present study. It is even unclear that the authors are trying to explain on the primary or secondary aims in the first paragraph of page 4. The authors should clearly mention the rationales of the primary and secondary aims in this paragraph. The authors should focus on mentioning only the purpose of the present study in the last paragraph of the Introduction section.

Our response #5: The rationale for the primary and secondary aims have been stated in the third (Page 3, Line 51 ~ Page 4, Line 59) and fourth (Page 4, Line 60-67) paragraphs, respectively. In addition, we have revised the last paragraph of the Introduction section to only refer to the purpose of the present study (Page 4, Line 68-70).

Comment #6: Page 4, line 71: General readers who are not familiar with meta-analysis may have no idea about the terms “scoping review” as well as “grey literature search”, or “umbrella review”. Please explain more courteously on them in the Materials and methods section or other appropriate sections.

Our response #6: For general readers unfamiliar with the principles of meta-analysis, descriptions of "scoping review” (Page 4, Line 73~), "grey literature search” (Page 5, Line 89~) and "umbrella review” (Page 5, Line 95~) have been provided in the Materials and methods.

Comment #7: Page 4, lines 60-61: The sentence is poorly written, and needs to be revised.

Our response #7: We have simplified the sentences to clarify their meaning (Page 4, Lines 62-67).

Comment #8: Page 5, lines 94-101: VO2max in the selected articles is directly measured by expired gas during maximal exercise test, or it is indirectly estimated by heart rate response during submaximal exercise test (e.g., using an equation of estimated maximum heart rate 220-age)? It is not clearly written whether the directly or indirectly measured VO2max or both are included (as VO2peak) in the present meta-analysis.

Our response #8: The present meta-analysis included data from both direct (measurement by breath gas analysis during the exercise test) and indirect methods (estimation using heart rate at several specific exercise intensities during sub-maximal exercise). We have provided the relevant information in the ‘Eligibility criteria’ section (Page 6, Lines 111-114).

Comment #9: Page 6, 102-110: Are these criteria are the criteria for Japanese population or other population or both? The authors performed a meta-analysis of other populations than Japanese population as a secondary aim (one of the major aims of the present study), but they mentioned “studies not in Japanese” as a criterion (line 106). Please be explicit not only about the eligibility criteria, but also the search strategy, study selection, data analysis for each of the Japanese population analysis and other population analysis.

Our response #9: We have presented the eligibility criteria (Page 6, Line 100~), search strategy (Page 5, Line 81~), study selection (Page 7, Line 126~) and data analysis (Page 8, Line 142~) for only the Japanese population in the Methods. To simplify the manuscript, this information on other populations has been presented in Supplemental materials (Supplemental materials, Page 3, Lines 41 ~ Page 5, Line 91).

Comment #10: Page 6, lines 109-110: The sentence is poorly written, and needs to be revised.

Our response #10: We have revised the sentence. The intention here was to inform that 'studies with duplicate data were excluded' and that in studies with duplicate data, more appropriate studies were selected in terms of sample size and study information (Page 7, Lines 122-123).

Comment #11: Page 6, lines 113-115: There is no description on the roles of the primary and secondary screening. There is also no explanation on the required conditions that articles pass the primary screening to the secondary screening. Please be explicit about that these primary and secondary screenings are used for Japanese population, other population, or both. In addition, for example, the authors mentioned that “The following items were extracted: (1) first author’s name, (2) year of publication, (3) sex, (4) age, (5) exercise mode, and (6) means and distributions of VO2peak and AT (SD, standard error, and confidence interval [CI])”. Please be explicit about that this (and other each procedure) is conducted as the primary screening or secondary screening.

Our response #11: We have revised the sentences as follows (Page 7, Lines 127-134): In primary screening, papers potentially containing information on the aerobic capacity of Japanese by sex and age were selected from titles and abstracts by two independent researchers (H.A. and M.M.). In the secondary screening, the same independent researchers perused the full text of the papers selected in the primary screening and selected those that precisely met the eligibility criteria. The papers handled during selection were managed using Mendeley Desktop (version 1.19.8) between the two researchers. From the papers selected in the secondary screening, data on (1) first author’s name, (2) year of publication, (3) sex, (4) age, (5) exercise mode, and (6) means and distributions (SD, standard error, or confidence interval [CI]) of VO2peak/kg and the value of AT/kg were extracted. In addition, information on the role and process of primary and secondary screening in other populations was revised in Supplemental material (Supplemental material, Page 4, Lines 62-75).

Comment #12: Page 7, line 127: VO2peak (or VO2max) is sometimes expressed as /kg lean body mass (LBM). The authors should clearly mention at least once that they used /kg body weight (not LBM) in the Materials and methods section (not only in footnotes of Tables and legends of Figures but also in the Methods section).

Our response #12: We have revised the sentences in the Methods (Page 7, Line 134). In addition, we have revised the footnotes of all Tables, Figure legends, and Supplemental material.

Comment #13: Page 7, lines 130-131: The reviewer could not understand the meaning of the sentence. Do you mean that the article (ref. 39) reported VO2peak and AT measured by indirect method? It is known that the VO2peak and AT measured by indirectly method are underestimated when they are compared with VO2peak and AT measured by directly method?

Our response #13: We have revised the sentences for more clarity (Page 8, Lines 143-149): We calculated VO2peak/kg (partly VO2max/kg) and AT/kg according to sex and age from the included studies by simple mean and SD to avoid the impact of studies with a large sample size. In particular, the sample size of a study by Kono et al (1997) was very large, accounting for 71% (55,521/78,714) of the total sample size in the present meta-analysis, which may affect the mean values and SD. The article (ref. 39) reported VO2peak using indirect methods.

Comment #14: Page 7, lines 132-134: The authors showed VO2peak and AT data by 10-year age group in Tables 1 and 2. However, they suddenly explain on the categorization of ≤19 years or ≥20 years, before they explain about the 10-year age group categorization first. It is also unclear that these data analyses are applied to Japanese population or other population, or both populations. Data analysis section is poorly written and should be drastically revised to prevent readers’ confusion.

Our response #14: To prevent reader confusion, the data analysis description has been drastically revised (Page 8, Lines 150-159). We have first stated that the estimated standard values for each aerobic capacity were stratified by sex and age, and that further stratification by age was performed by 10-year age group categories in the data analysis. Then, we have described that the effect of ageing on each aerobic capacity (in both Japanese and other populations) was stratified into two categories based on Japanese adult norms (≤19 years or ≥20 years) and the data was analyzed.

Comment #15: Page 7, lines 135-136: The authors mentioned “The slopes and intercepts of the mean and age for each indicator from each study were determined using…”. Is it correct? It should be “The slopes and intercepts of the association of age with VO2peak and AT for each age category (≦19 years, ≧20 years) were determined using…”.

Our response #15: We appreciate the reviewer for the suggestion. We have revised the sentence accordingly (Page 8, Lines 154-156).

Comment #16: Page 8, line 142: Please be explicit about that these are accepted articles and data extraction for Japanese population, or other population than Japanese, or both.

Our response #16: We have clearly stated “Accepted article and data extraction for Japanese population” in the Manuscript (Page 9, Line 165~) and “Accepted article and data extraction for other populations” in Supplemental material (Supplemental material, Page 5, Line 94~).

Comment #17: Page 10, Table 1: The authors may touch the possible reason why the VO2peak data measured by run are lower than VO2peak data measured by cycle in the age groups of 4-9 and 10-19 in the Materials and methods section or Discussion section.

Our response #17: We have provided the following explanation in the first paragraph of the Discussion section under 'Differences in exercise mode': “In contrast, VO2peak/kg was lower for running than for cycling in Japanese men and women aged 4–9 and 10–19 years. Although the detailed reasons for this observation is beyond the scope of this review, it may be due to sampling bias rather than the physiological effects of the different exercises. Additionally, the reversal phenomenon of cycling and running may presumably be due to the small number of studies on the aerobic capacity of minors, and the inclusion of sports children in the studies using cycling (Tamiya, 1991) [34]” (Page 19, Line 338-346).

Comment #18: Page 10, Table 1: The percentage of AT in VO2peak is approximately 50% in the age groups of young and middle-aged men (Table 1) and women (Table 2), but the percentage is apparently higher in older men and women (55-61% in Tables 1 and 2). To the reviewer’s knowledge, endurance athletes have a higher percentage of AT in VO2peak (>60%), but not in older people. How do the authors explain this phenomenon?

Our response #18: As per the reviewer’s comment, we have described the reasons (Page 18, Line 308-326).

Comment #19: Page 13, lines 216-220: It is unclear that the VO2peak and AT data shown in S2 and S3 figures are measured by cycle, run, or combined with cycle and run.

Our response #19: Figures S2 and S3 show the combined running and cycling data. This has been described in the Manuscript (Page 10, Line 192; Page 14, Line 231; Page 32, Line 741; Page 33, Line 754) and Supplemental material (Supplemental material, Page 6, Lines 110-111; Page 7, Lines 126-127; Page 22, Line 204; Page 23, Line 216).

Comment #20: Page 16, lines 273-279: Based on the AT data in the other age groups than 70–89-year group, it seems that the decrease in AT in Japanese population compared with other population (shown in S3 figure) is greater than the decrease in VO2peak in the Japanese population compared with other population (shown in S2 figure). How do the authors explain this interesting phenomenon?

Our response #20: We appreciate the reviewer for the constructive suggestions. We have added the relevant sentences as follows (Page 18, Line 327 ~ Page 19, Line 335). “Interestingly, beyond the differences in VO2peak (S2 Fig), AT in Japanese men aged 20–69 years was markedly lower than that in other populations (S3 Fig). The mechanisms contributing to this phenomenon are unknown. However, given that AT largely depends on the skeletal muscle’s oxidative capacity, the skeletal muscle mass or skeletal muscle’s oxidative capacity may differ between Japanese and other populations. Silva et al. (2010) reported that skeletal muscle mass decreases after 27 years of age, and that ethnic differences exist in this ageing phenomenon of skeletal muscle mass. They also reported that skeletal muscle mass is lowest in Asians, including Japanese; however, only women were included in their study [84]. Further research should be conducted on the mechanisms underlying AT differences between ethnic groups with data from both sexes.”

Comment #21: Page 18, line 310: The authors stated that “identifying the distribution of aerobic capacity in population may help estimate their health status and develop a health promotion strategy” in Abstract. In addition, they mentioned that “estimated standard values may help set or update more practical reference values for health promotion in the Japanese population” in the conclusion. However, this reviewer could not clearly understand the reason why the estimated standard values may help set or update more practical reference values for health promotion in the Japanese population. Please describe more how the standard values of VO2peak and AT estimated by the present study actually contribute to the development of the reference value of the aerobic capacities for future health promotion, in this Perspectives section or other appropriate sections, more clearly and more concretely.

Our response #21: We have described in the first paragraph of the Perspectives (Page 21, Line 372~)” and in the Conclusions (Page 22, Line 393~) how the estimated standard values can help set and update the reference values. The introduction has also been revised to reflect this (Page 3, Line 54~). 

Responses to Reviewer 2’s comments

Comment #1: Line 281. VO2peak/kg is lower in Run than in Cycle for both males and females in their teens. Perhaps this is a discrepancy arising from Tamiya's (1991) higher results for teenagers with Cycle. The author should add a discussion of this result. (No need to re-review.)

Our response #1: We appreciate he reviewer for the constructive suggestions. We have provided the following explanation in the first paragraph of the Discussion section under 'Differences in exercise mode': “In contrast, VO2peak/kg was lower for running than for cycling in Japanese men and women aged 4–9 and 10–19 years. Although the detailed reasons for this observation is beyond the scope of this review, it may be due to sampling bias rather than the physiological effects of the different exercises. Additionally, the reversal phenomenon of cycling and running may presumably be due to the small number of studies on the aerobic capacity of minors, and the inclusion of sports children in the studies using cycling (Tamiya, 1991) [34]” (Page 19, Line 338-346).

---

## [Decision Letter · Decision Letter 1]

30 Aug 2023

PONE-D-23-15750R1Estimated standard values of aerobic capacity according to sex and age in a Japanese population: a scoping reviewPLOS ONE

Dear Dr. Miyachi,

Thank you for submitting your manuscript to PLOS ONE. After careful consideration, we feel that it has merit but does not fully meet PLOS ONE’s publication criteria as it currently stands. Therefore, we invite you to submit a revised version of the manuscript that addresses the points raised during the review process.

We look forward to receiving your revised manuscript.

Kind regards,

Yosuke Yamada

Academic Editor

PLOS ONE

Journal Requirements:

Reviewers' comments:

Reviewer's Responses to Questions

**Comments to the Author**

1. If the authors have adequately addressed your comments raised in a previous round of review and you feel that this manuscript is now acceptable for publication, you may indicate that here to bypass the “Comments to the Author” section, enter your conflict of interest statement in the “Confidential to Editor” section, and submit your "Accept" recommendation.

Reviewer #1: (No Response)

Reviewer #2: All comments have been addressed

2. Is the manuscript technically sound, and do the data support the conclusions?

Reviewer #1: Yes

Reviewer #2: (No Response)

3. Has the statistical analysis been performed appropriately and rigorously? 

Reviewer #1: Yes

Reviewer #2: (No Response)

4. Have the authors made all data underlying the findings in their manuscript fully available?

Reviewer #1: Yes

Reviewer #2: (No Response)

5. Is the manuscript presented in an intelligible fashion and written in standard English?

Reviewer #1: Yes

Reviewer #2: (No Response)

6. Review Comments to the Author

Reviewer #1: The authors suitably responded to most of the reviewer’s comments. The reviewer believes that the manuscript has been much improved. However, unfortunately, the authors failed in the revision of the Abstract. An abstract is most important, as is an article title.

The authors responded, “However, this information has been only briefly provided to avoid confusion among readers regarding the primary purpose of this meta-analysis, which was to obtain a standard value estimated of aerobic capacity in the Japanese population (Page 2, Lines 20, 23-24).” If so, they should clearly mention that the comparison of the estimated standard values of the Japanese with those of other populations was performed “as a secondary or supplementary analysis” in the Abstract. Why don’t they simply express so? A brief description does not necessarily mean a secondary description, and it is just a lack of sufficient explanation. The authors also mention that “This study aimed to estimate standard values of aerobic capacity (peak oxygen uptake [VO2peak]/kg and anaerobic threshold [AT]/kg) for the Japanese population stratified by sex and age using a meta-analysis and to compare them with those of other populations” in Abstract. This sentence may also confuse readers, because readers may misunderstand that comparing aerobic capacity with those of other populations is one of the two primary analyses.

Reviewer #2: (No Response)

7. PLOS authors have the option to publish the peer review history of their article (what does this mean?). If published, this will include your full peer review and any attached files.

Reviewer #1: No

Reviewer #2: No

---

## [Author Response · Author response to Decision Letter 1]

31 Aug 2023

We would like to thank the editor and reviewers for reading the manuscript and providing their constructive comments. Our responses to the reviewers are listed in "Response to Reviewers" file and the corresponding corrections have been made in the "Revised Manuscript with Track Changes". The reviewers' comments (in black) are listed below, followed by our responses (in red).

Responses to Editor’s comments

Comment: Please review your reference list to ensure that it is complete and correct. If you have cited papers that have been retracted, please include the rationale for doing so in the manuscript text, or remove these references and replace them with relevant current references. Any changes to the reference list should be mentioned in the rebuttal letter that accompanies your revised manuscript. If you need to cite a retracted article, indicate the article’s retracted status in the References list and also include a citation and full reference for the retraction notice.

Our response: We checked the reference list according to the editor's comments. The reference list does not include retracted papers. However, there were some cited references in Japanese papers that did not comply with PLOS ONE's submission guideline and ICMJE sample references. Therefore, we modified these cited references according to these guidelines.

Responses to Reviewer 1’s comments

Comment #1: The authors suitably responded to most of the reviewer’s comments. The reviewer believes that the manuscript has been much improved. However, unfortunately, the authors failed in the revision of the Abstract. An abstract is most important, as is an article title.

Our response #1: We appreciate the constructive comments of the reviewer and have revised the manuscript accordingly. We revised the abstract, which is as important as the article title. We hope the revised abstract will help readers better understand the subject.

Comment #2: The authors responded, “However, this information has been only briefly provided to avoid confusion among readers regarding the primary purpose of this meta-analysis, which was to obtain a standard value estimated of aerobic capacity in the Japanese population (Page 2, Lines 20, 23-24).” If so, they should clearly mention that the comparison of the estimated standard values of the Japanese with those of other populations was performed “as a secondary or supplementary analysis” in the Abstract. Why don’t they simply express so? A brief description does not necessarily mean a secondary description, and it is just a lack of sufficient explanation. The authors also mention that “This study aimed to estimate standard values of aerobic capacity (peak oxygen uptake [VO2peak]/kg and anaerobic threshold [AT]/kg) for the Japanese population stratified by sex and age using a meta-analysis and to compare them with those of other populations” in Abstract. This sentence may also confuse readers, because readers may misunderstand that comparing aerobic capacity with those of other populations is one of the two primary analyses. 

Our response #2: As per the reviewer’s comment, we recognized that the revised abstracts were not well explained and adequately expressed. To avoid confusion among readers, we have revised that the comparison of estimated standard values between the Japanese and other populations was conducted 'as a supplementary analysis'. The revised sentence is: ‘Moreover, a comparison of the estimated standard values of the Japanese with those of other populations was performed as a supplementary analysis’ (Page 2, Lines 20-21).

---

## [Decision Letter · Decision Letter 2]

4 Sep 2023

Estimated standard values of aerobic capacity according to sex and age in a Japanese population: a scoping review

PONE-D-23-15750R2

Dear Dr. Miyachi,

We’re pleased to inform you that your manuscript has been judged scientifically suitable for publication and will be formally accepted for publication once it meets all outstanding technical requirements.

Kind regards,

Yosuke Yamada

Academic Editor

PLOS ONE

Additional Editor Comments (optional):

Congratulations!

Reviewers' comments:

Reviewer's Responses to Questions

**Comments to the Author**

1. If the authors have adequately addressed your comments raised in a previous round of review and you feel that this manuscript is now acceptable for publication, you may indicate that here to bypass the “Comments to the Author” section, enter your conflict of interest statement in the “Confidential to Editor” section, and submit your "Accept" recommendation.

Reviewer #1: All comments have been addressed

2. Is the manuscript technically sound, and do the data support the conclusions?

Reviewer #1: Yes

3. Has the statistical analysis been performed appropriately and rigorously? 

Reviewer #1: Yes

4. Have the authors made all data underlying the findings in their manuscript fully available?

Reviewer #1: Yes

5. Is the manuscript presented in an intelligible fashion and written in standard English?

Reviewer #1: Yes

6. Review Comments to the Author

Reviewer #1: (No Response)

7. PLOS authors have the option to publish the peer review history of their article (what does this mean?). If published, this will include your full peer review and any attached files.

Reviewer #1: No

---

## [Editor Report · Acceptance letter]

7 Sep 2023

PONE-D-23-15750R2 

Estimated standard values of aerobic capacity according to sex and age in a Japanese population: a scoping review 

Dear Dr. Miyachi:

I'm pleased to inform you that your manuscript has been deemed suitable for publication in PLOS ONE. Congratulations! Your manuscript is now with our production department. 

Kind regards, 

on behalf of

Dr. Yosuke Yamada 

Academic Editor

PLOS ONE